# Combined Aerobic Exercise Training and *Chlorella* Intake Reduces Arterial Stiffness through Enhanced Arterial Nitric Oxide Production in Obese Rats

**DOI:** 10.3390/nu16183080

**Published:** 2024-09-13

**Authors:** Henry Yamazaki, Shumpei Fujie, Kenichiro Inoue, Masataka Uchida, Motoyuki Iemitsu

**Affiliations:** Faculty of Sport and Health Science, Ritsumeikan University, Kusatsu 525-8577, Japan; sh0256if@ed.ritsumei.ac.jp (H.Y.); sh0118es@ed.ritsumei.ac.jp (K.I.); m-uchida@fc.ritsumei.ac.jp (M.U.); iemitsu@fc.ritsumei.ac.jp (M.I.)

**Keywords:** obesity, arterial stiffness, nitric oxide, *Chlorella*, aerobic exercise

## Abstract

This study aimed to assess the effect of a combination of aerobic exercise training (ET) and *Chlorella* (CH) intake on arterial nitric oxide (NO) production and arterial stiffness in obese rats. Twenty-week-old obese male rats were randomly grouped into four (*n* = 6): OBESE-SED (sedentary control), OBESE-ET (treadmill 25 m/min, 1 h, 5 d/week), OBESE-CH (0.5% *Chlorella* powder in normal diet), and OBESE-ET+CH (combination of ET and CH intake) groups. The carotid–femoral pulse wave velocity (cfPWV), an index of arterial stiffness, was significantly lesser in the OBESE-ET, OBESE-CH, and OBESE-ET+CH groups than in the OBESE-SED group, and in the OBESE-ET+CH group significantly further enhanced these effects compared with the OBESE-ET and OBESE-CH groups. Additionally, arterial nitrate/nitrite (NOx) levels were significantly greater in the OBESE-ET, OBESE-CH, and OBESE-ET+CH groups than in the OBESE-SED group, and the OBESE-ET+CH group compared with the OBESE-ET and OBESE-CH groups. Furthermore, arterial NOx levels were positively correlated with arterial endothelial NO synthase phosphorylation levels (r = 0.489, *p <* 0.05) and negatively correlated with cfPWV (r = −0.568, *p <* 0.05). In conclusion, a combination of ET and CH intake may reduce arterial stiffness via an enhancement of the arterial NO signaling pathway in obese rats.

## 1. Introduction

The global prevalence of overweight adults and patients with obesity is high and increasing [1,2]. Obesity is a major lifestyle illness that leads to further health concerns and induces numerous chronic diseases [3]. It is generally accepted that obesity is associated with vascular endothelial dysfunction, leading to an increase in arterial stiffness [4]. For cardiovascular diseases and a predictor of cardiovascular events, an increase in arterial stiffness is an independent risk factor [5]. Arterial stiffness is regulated by the production of vascular endothelium-derived relaxing factors, such as nitric oxide (NO), through activation of the protein kinase b (Akt)/endothelial NO synthase (eNOS) signaling pathway in endothelial cells [6]. In patients with obesity, a decrease in the bioavailability of NO is involved in the increase in arterial stiffness [4]. Aerobic exercise training in patients with obesity decreases arterial stiffness through an enhancement of arterial NO production, with activation of Akt/eNOS signaling pathway, in addition to reducing fat accumulation [7].

*Chlorella* is a unicellular freshwater microalga and a food supplement because of its various nutrients, including multiple amino acids, dietary fiber, vitamins, and minerals [8]. It has been shown to have several beneficial effects. Chronic *Chlorella* supplementation accelerates immune function [9], improves obesity and blood lipid profiles [10], and reduces insulin resistance [11]. Furthermore, new effects of *Chlorella* intake have also been reported, such as lowering brachial–ankle pulse wave velocity (baPWV), an indicator of arterial stiffness, and increasing plasma nitrate/nitrite (NOx, an index of NO production) levels in middle-aged and older adults [12]. Our recent study showed, in senescence-accelerated mice, chronic feeding with a chow diet containing *Chlorella* improved the endothelium-dependent vasorelaxation response, with enhanced NO production through the prompting of arterial Akt/eNOS signaling pathway [13]. However, it is uncertain whether habitual *Chlorella* intake combined with aerobic exercise training further reduces arterial stiffness through an enhancement of arterial NO production in the context of obesity.

Therefore, the purpose of this research was to clarify whether habitual *Chlorella* intake combined with aerobic exercise training further reduces arterial stiffness through an enhancement of arterial NO production via the activation of the Akt/eNOS signaling pathway in rats with obesity.

## 2. Materials and Methods

### 2.1. Animals and Protocol

Male Otsuka Long–Evans Tokushima fatty (OLETF) rats at six weeks old obtained from Japan SLC, Inc. (Shizuoka, Japan) were used as the model of obesity. Ethical approval for the study was acquired from the Committee on Animal Care at Ritsumeikan University in Japan and animal care was conducted in accordance with the Guiding Principles for the Care and Use of Animals, based on the Declaration of Helsinki (Helsinki, Finland). Under managed environments (12/12-h light/dark cycle), all rats were housed individually in the animal facility. After 14 weeks, the male OLETF rats at 20-week-old were randomly divided into four groups (*n* = 6 per group): sedentary control (OBESE-SED), aerobic exercise training (OBESE-ET), *Chlorella* intake (OBESE-CH), and a combination of aerobic exercise training and *Chlorella* intake (OBESE-ET+CH) groups. During the 8-week experimental period, the OBESE-SED and OBESE-ET groups were given unrestricted access to water and a regular diet (CE-2; CLEA Japan, Tokyo, Japan) *ad libitum*. The OBESE-CH and OBESE-ET+CH groups consumed the same food supplemented with 0.5% *Chlorella* powder (Sun Chlorella Corp., Kyoto, Japan) as previously described [13]. Additionally, healthy, non-diabetic, age-equivalent Long–Evans Tokushima Otsuka rats (*n* = 6) were considered as a healthy sedentary control (Healthy) group. After each 8-week intervention, we evaluated body weight; systolic (SBP) and diastolic (DBP) blood pressures; and carotid–femoral pulse wave velocity (cfPWV), as an index of central arterial stiffness, during fasting. We extracted blood samples from the abdominal aorta and analyzed blood glucose levels under general anesthesia. Following a sacrifice, the soleus muscles, epididymal fat, and abdominal aorta were expeditiously resected, cleaned in ice-cold saline, weighed, frozen using liquid nitrogen, and preserved at −80 °C for further analysis.

### 2.2. Carotid–Femoral Pulse Wave Velocity and Blood Pressures

The carotid–femoral pulse wave velocity (cfPWV), heart rate (HR), systolic blood pressure (SBP), and diastolic blood pressure (DBP) were assessed as previously described [7]. The cfPWV and BPs were assessed using two catheters. The cfPWV was derived by dividing the distance of propagation, the straight-line distance between the catheter tips, by the propagation time, and SBP and DBP were simultaneously measured.

### 2.3. Aerobic Exercise Training Protocol

The OBESE-ET and OBESE-ET+CH groups underwent training on a small animal treadmill at 10 to 15 m/min for 3 d to warm up before the training experimental period. The rats in the OBESE-ET and OBESE-ET+CH groups completed a 1-h treadmill session at 25 m per min, at a flat incline, 5 days a week for 8 weeks, as previously described [7].

### 2.4. Citrate Synthase Activity

Citrate synthase (CS) activity was measured as an indicator of adaptation to aerobic exercise training [7] using soleus muscles, which are recruited during treadmill running exercises.

### 2.5. Western Blot Analysis

Western blotting was used to evaluate eNOS phosphorylation and Akt phosphorylation, as previously described [7]. In short, to homogenize tissue samples, frozen tissues were cut and placed in a round bottom tube with beads, followed by the addition of 300 μL of Radio-Immunoprecipitation Assay (RIPA) buffer and rotation for 20 min × 4 times. After centrifugation at 130 × 100 rpm at 4 °C for 30 min, the supernatant was collected from the tube and used as a sample. For each sample, a total protein amount of 20 μg was prepared. The aorta was separated by 10% sodium dodecyl sulfate polyacrylamide gel and transferred to polyvinylidene difluoride membranes (Millipore, Billerica, MA, USA). Blocking buffer was applied to the membranes for 2 h (1% skim milk in phosphate-buffered saline with 0.1% Tween 20 [PBS-T]), followed by a 12-h incubation at 4 °C in blocking buffer with antibodies (diluted 1:500 in blocking buffer) against eNOS phosphorylated on Ser1177 (ab184154; Abcam, Piscataway, NJ, USA), total eNOS (5329915; BD Biosciences, Franklin, NJ, USA), Akt phosphorylated on Ser473 (#9271, Cell Signaling Technology, Danvers, MA, USA), or total Akt (#9272, Cell Signaling Technology). The membranes underwent three washes with PBS-T, followed by a 1 h incubation at room temperature (22–24 °C) with horseradish-peroxidase-conjugated secondary antibody and anti-rabbit (Cell Signaling Technology) or anti-mouse (GE Healthcare, UK Ltd., Buckinghamshire, UK) immunoglobulin diluted 1:3000 in blocking buffer. Finally, phosphorylated and total eNOS and Akt levels were detected using the Enhanced Chemiluminescence Plus system (GE Healthcare) and visualized on a FUSION FX instrument (Vilber Lourmat, Collégien, France). ImageJ software was used to perform densitometry (ver 1.48; National Institutes of Health, Bethesda, MD, USA).

### 2.6. Griess Assay

Arterial nitrate/nitrite (NOx) concentrations were assessed with the Griess assay (R&D Systems, Minneapolis, MN, USA), as previously described [7].

### 2.7. Statistical Analysis

Results are reported as the mean ± standard error. Statistical evaluations were performed using a one-way analysis of variance (ANOVA). A post-hoc comparison test was used to correct for multiple comparisons (Fisher’s test) when ANOVA results indicated significant differences. Pearson’s correlation coefficients were used to determine the correlations between arterial NOx levels and the arterial levels of phosphorylated eNOS and cfPWV. Results were considered statistically significant if *p* < 0.05. All statistical analyses were performed using Stat View (5.0; SAS Institute, Tokyo, Japan).

## 3. Results

### Animal Characteristics

Body weight and epididymal fat mass were significantly greater in the OBESE-SED group than in the Healthy group (*p* < 0.05, Table 1). Body weight was significantly lesser in the OBESE-ET group and epididymal fat mass was significantly lesser in the OBESE-ET and OBESE-ET+CH groups than in the OBESE-SED group (*p* < 0.05, Table 1). Body weight in the OBESE-CH and OBESE-ET+CH groups were significantly greater than those Healthy and OBESE-ET groups (*p* < 0.05, Table 1). Soleus muscle mass was significantly lesser in the OBESE-ET+CH group than in the Healthy and OBESE-ET groups (*p* < 0.05, Table 1), but was significantly greater in the OBESE-ET group than the OBESE-SED group (*p* < 0.05, Table 1). Soleus CS enzyme activity was significantly greater in the OBESE-ET and OBESE-ET+CH groups than in the Healthy and OBESE-SED groups (*p* < 0.05, Table 1). Fasting blood glucose levels were significantly greater in the OBESE-SED, OBESE-ET, and OBESE-CH groups than the Healthy group (*p* < 0.05, Table 1); significantly lesser in the OBESE-ET, OBESE-CH, and OBESE-ET+CH groups than the OBESE-SED group (*p* < 0.05, Table 1); and significantly lesser in the OBESE-ET and OBESE-ET+CH groups than the OBESE-CH group (*p* < 0.05, Table 1). HOMA-IR scores were significantly greater in the OBESE-SED, OBESE-ET, OBESE-CH, and OBESE-ET+CH groups than the Healthy group (*p* < 0.05, Table 1); significantly lesser in the OBESE-ET, OBESE-CH, and OBESE-ET+CH groups than the OBESE-SED group (*p* < 0.05, Table 1); significantly lesser, in the OBESE-ET and OBESE-ET+CH groups than the OBESE-CH group (*p* < 0.05, Table 1); and significantly lesser in the OBESE-ET+CH group than the OBESE-ET group (*p* < 0.05, Table 1). No significant differences in HR, SBP, or DBP were observed between groups. Average food intake was significantly higher in the OBESE-SED, OBESE-CH, and OBESE-ET+CH groups than in the Healthy and OBESE-ET groups (*p* < 0.05, Table 1). However, no significant differences in average food intake were observed among the OBESE-SED, OBESE-CH, and OBESE-ET+CH groups (Table 1).

The cfPWV was significantly greater in the Healthy group than the OBESE-SED group (*p* < 0.05, Table 1 and Figure 1), but significantly lesser in the OBESE-CH, OBESE-ET, and OBESE-ET+CH groups than the OBESE-SED group (*p* < 0.05, Table 1 and Figure 1). No significant difference in the cfPWV was observed between the OBESE-CH and OBESE-ET groups and was significantly greater in the OBESE-ET+CH group than the OBESE-CH and OBESE-ET groups (*p* < 0.05, Table 1 and Figure 1).

Arterial Akt phosphorylation levels were significantly lesser in the OBESE-SED group than the Healthy group (*p* < 0.05, Table 1 and Figure 2A), but were significantly greater in the OBESE-ET+CH group than the OBESE-SED and OBESE-ET groups (*p* < 0.05, Table 1 and Figure 2A). Arterial eNOS phosphorylation levels were significantly lesser in the OBESE-SED and OBESE-CH groups than the Healthy group (*p* < 0.05, Table 1 and Figure 2B), but were significantly greater in the OBESE-ET and OBESE-ET+CH groups than the OBESE-SED and OBESE-CH groups (*p* < 0.05, Table 1 and Figure 2B).

Arterial NOx levels were significantly lesser in the OBESE-SED group than the Healthy group (*p* < 0.05, Table 1 and Figure 3), but were significantly greater in the OBESE-CH, OBESE-ET, and OBESE-ET+CH groups than the OBESE-SED group. No significant difference in arterial NOx levels was observed between the OBESE-CH and OBESE-ET groups, but they were significantly greater in the OBESE-ET+CH group than the OBESE-CH and OBESE-ET groups (*p* < 0.05, Table 1 and Figure 3).

There were significant differences in the body weight, epididymal fat mass, fasting blood glucose levels, HOMA-IR score, arterial eNOS phosphorylation levels, arterial NOx levels, and average food intake between the Healthy and OBESE-SED groups (*p* < 0.05, Table 2). Additionally, there were significant differences in the body weight, epididymal fat mass, soleus CS enzyme activity, fasting blood glucose levels, HOMA-IR score, arterial eNOS phosphorylation levels, arterial NOx levels, and average food intake between the OBESE-ET and OBESE-SED groups (*p* < 0.05, Table 2). In addition, there were significant differences in the fasting blood glucose levels, HOMA-IR score, cfPWV, arterial Akt phosphorylation levels, and arterial NOx levels between the OBESE-CH and OBESE-SED groups (*p* < 0.05, Table 2). Furthermore, there were significant differences in the epididymal fat mass, soleus CS enzyme activity, fasting blood glucose levels, HOMA-IR score, cfPWV, arterial Akt phosphorylation levels, arterial eNOS phosphorylation levels, and arterial NOx levels between the OBESE-ET+CH and OBESE-SED group (*p* < 0.05, Table 2).

The arterial NOx levels were positively correlated with the arterial eNOS phosphorylation levels (*p* < 0.05, r = 0.489, Figure 4A), and negatively correlated with the cfPWV (*p* < 0.05, r = −0.568, Figure 4B).

## 4. Discussion

In this study, we found that a combination of aerobic exercise training and *Chlorella* intake further reduced arterial stiffness through an enhancement of arterial NO production, as compared with aerobic exercise training or *Chlorella* intake alone, in obese rats. Moreover, this combination further increased arterial NOx levels and arterial Akt and eNOS phosphorylation levels. Furthermore, arterial eNOS phosphorylation levels were positively correlated with arterial NOx levels. Therefore, these data indicate that a combination of aerobic exercise training and *Chlorella* intake may further reduce arterial stiffness through an enhancement of arterial NO production by activating the Akt/eNOS signaling pathway in obese rats.

Previous research showed that chronic *Chlorella* intake decreased the baPWV in middle-aged and older individuals [12]. In addition, aerobic exercise training in adults with obesity significantly reduced the cfPWV [7]. However, it is still unknown whether habitual *Chlorella* intake combined with aerobic exercise training further reduces arterial stiffness. The current research showed that, compared with chronic aerobic exercise training or *Chlorella* intake alone, the combination of aerobic exercise training and *Chlorella* intake was found to significantly decrease the cfPWV in rats with obesity.

In our recent previous research, we found that aerobic exercise training increased the circulating levels of NOx, and the phosphorylation levels of arterial Akt and eNOS in rats with obesity [7]. Furthermore, chronic *Chlorella* intake in aged mice was found to increase arterial NO production through the elevation of arterial Akt/eNOS signaling pathway, and combining aerobic exercise training and *Chlorella* intake further enhanced these effects [13]. Moreover, aerobic exercise training or *Chlorella* intake alone, and combined aerobic exercise training and *Chlorella* intake can induce acetylcholine-induced vasorelaxation, and consequently, improve the endothelial function of the aorta [13]. However, it is unclear whether habitual aerobic exercise training combined with *Chlorella* intake further increases arterial NO production through prompting of the Akt/eNOS signaling pathway in rats with obesity. In the present study, compared with aerobic exercise training and *Chlorella* intake alone, the combination of aerobic exercise training and *Chlorella* intake was found to significantly increase arterial NOx levels and phosphorylated eNOS levels in rats with obesity. Furthermore, arterial NOx levels were positively correlated with phosphorylated eNOS levels and negatively correlated with the cfPWV. Thus, the combination of aerobic exercise training and *Chlorella* intake in rats with obesity might further reduce arterial stiffness via an enhancement of arterial NO production.

We found that chronic *Chlorella* intake led to an enhancement of arterial NO production in obese rats. However, it is uncertain what nutrients including *Chlorella* increased arterial NO production. *Chlorella* is a multi-nutrient supplement that contains amino acids and vitamins that affect vasorelaxation via the acceleration of arterial NO production. L-arginine, a substrate of eNOS, is a precursor of NO in the vascular endothelium [14]. A previous study showed that dietary L-arginine supplementation for 10 weeks significantly improved endothelium-dependent relaxation in hypercholesterolemic rabbits [15]. Furthermore, the mean atherosclerotic lesion area was significantly decreased in cholesterol-fed mice administered L-arginine [16]. Mice that lack endothelial vitamin D receptor expression have reduced bioavailability of NO caused by a reduction in aortic eNOS mRNA expression levels [17]. In addition, the active hormone form of vitamin D, 1α,25-dihydroxyvitamin D, increases the phosphorylation levels of Akt and eNOS, leading to augmented endothelial NO production [18]. Thus, vitamin D may promote NO bioavailability through the activation of arterial eNOS. Moreover, vitamin C may enhance NO bioavailability by increasing the stability of BH_4_, which is a co-factor of eNOS [19]. In addition, 4 weeks of therapy with vitamin E has been shown to stimulate NO-related endothelial function in hypercholesterolemic individuals [20]. As we did not examine each of the nutritional components of *Chlorella*, future studies are required to determine the specific nutrients in *Chlorella* that increased arterial NO production and caused beneficial effects on endothelial function, resulting in a decrease in arterial stiffness. Another limitation of this study was the lack of measurements of other NOS isoforms in several tissues, including the artery. Analyses of all three NOS isoforms in several tissues could provide insight into the mechanisms underlying the *Chlorella* intake-induced increase in arterial NOx levels in obese rats.

In the current study, habitual aerobic exercise training combined with *Chlorella* intake further increased arterial NOx levels and phosphorylated eNOS and Akt levels, leading to a decrease in the cfPWV. Furthermore, chronic *Chlorella* intake alone also has beneficial effects on the arterial stiffness response, with increased arterial NO production. Therefore, chronic *Chlorella* intake may be useful for patients with obesity, because it is hard for such patients to conduct chronic aerobic exercise training due to motor disorders [21]. Additionally, combination training induces a greater improvement in arterial stiffness than aerobic exercise training or *Chlorella* intake alone. Therefore, the combination of habitual aerobic exercise training and *Chlorella* intake might be a more effective treatment for patients with obesity.

## 5. Conclusions

The findings of this research suggest that habitual aerobic exercise training combined with *Chlorella* intake further reduced arterial stiffness, compared with *Chlorella* intake or aerobic exercise training alone, through an enhancement of arterial NO production by activated arterial Akt/eNOS signaling pathway in rats with obesity.

## Figures and Tables

**Figure 1 nutrients-16-03080-f001:**
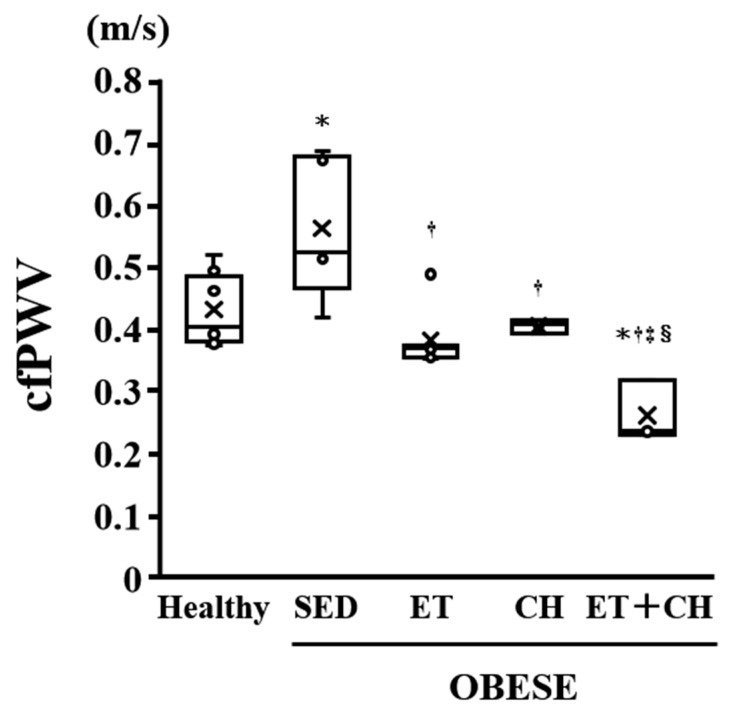
Comparison of carotid–femoral pulse wave velocity (cfPWV) in the Healthy, OBESE-SED, OBESE-ET, OBESE-CH, and OBESE-ET+CH groups. Data are expressed as the mean ± standard deviation. X represents the mean value. Circle represents the individual data. * *p* < 0.05 vs. the Healthy group, † *p* < 0.05 vs. the OBESE-SED group, ‡ *p* < 0.05 vs. the OBESE-CH group, § *p* < 0.05 vs. the OBESE-ET group.

**Figure 2 nutrients-16-03080-f002:**
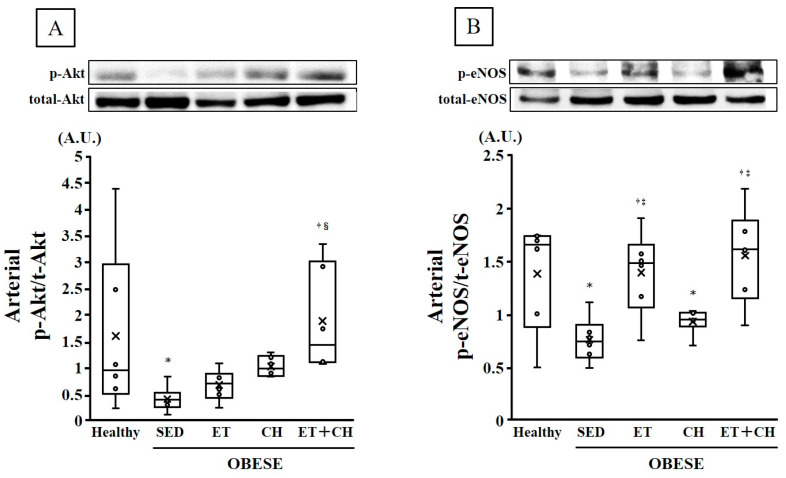
Comparison of arterial phosphorylated-Akt (*p*-Akt)/total-Akt (t-Akt) (**A**) and phosphorylated-eNOS (*p*-eNOS)/total-eNOS (t-eNOS) (**B**) ratios in the Healthy, OBESE-SED, OBESE-ET OBESE-CH, and OBESE-ET+CH groups. Data are expressed as the mean ± standard deviation. X represents the mean value. Circle represents the individual data. * *p* < 0.05 vs. the Healthy group, † *p* < 0.05 vs. the OBESE-SED group, ‡ *p* < 0.05 vs. the OBESE-CH group, § *p* < 0.05 vs. the OBESE-ET group.

**Figure 3 nutrients-16-03080-f003:**
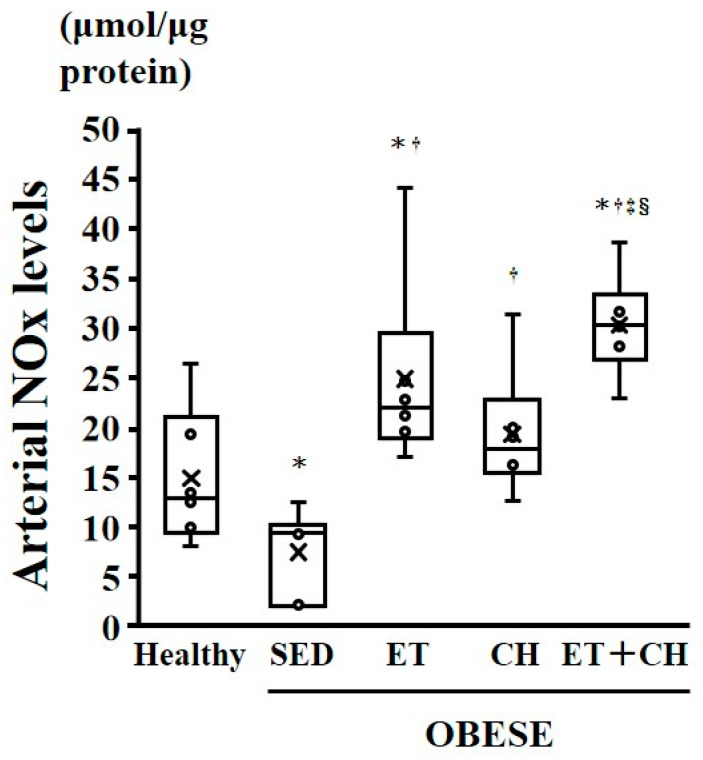
Comparison of arterial nitrate/nitrite (NOx) concentrations in the Healthy, OBESE-SED, OBESE-ET OBESE-CH, and OBESE-ET+CH groups. Data are expressed as mean ± standard deviation. X represents the mean value. Circle represents the individual data. * *p* < 0.05 vs. the Healthy group, † *p* < 0.05 vs. the OBESE-SED group, ‡ *p* < 0.05 vs. the OBESE-CH group, § *p* < 0.05 vs. the OBESE-ET group.

**Figure 4 nutrients-16-03080-f004:**
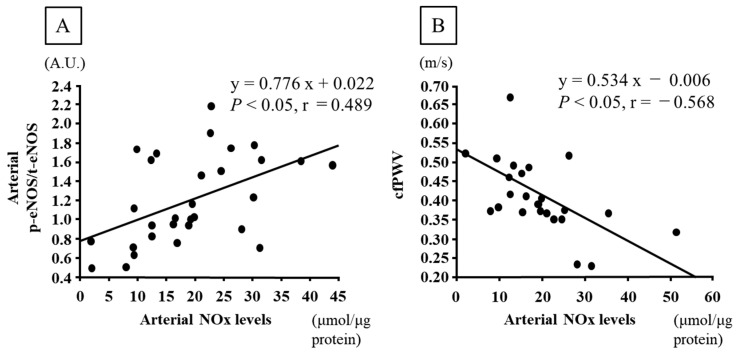
Correlations between arterial nitrate/nitrite (NOx) levels and arterial eNOS phosphorylation levels (**A**) and carotid–femoral pulse wave velocity (cfPWV) (**B**).

**Table 1 nutrients-16-03080-t001:** Animal characteristics.

	Healthy (*n* = 6)	OBESE			
		SED (*n* = 6)	ET (*n* = 6)	CH (*n* = 6)	ET+CH (*n* = 6)
Body weight (g)	481.8 ± 26.3	580.7 ± 36.6 *	490.6 ± 35.3 ^†^	566.7 ± 25.1 *^§^	570.0 ± 33.0 *^§^
Epididymal fat mass (g)	7.35 ± 1.50	11.01 ± 0.70 *	7.33 ± 1.94 ^†^	9.12 ± 2.86	7.63 ± 1.16 ^†^
Soleus muscle mass (g)	0.42 ± 0.05	0.37 ± 0.07	0.44 ± 0.05 ^†‡^	0.38 ± 0.05	0.36 ± 0.03 *^§^
Soleus CS enzyme activity (µmol/g/min)	13.19 ± 9.43	11.93 ± 2.39	22.40 ± 10.57 *^†^	15.68 ± 8.68	24.65 ± 3.03 *^†^
Fasting blood glucose levels (mmol/L)	5.71 ± 0.51	19.19 ± 2.03 *	8.46 ± 1.67 *^†‡^	16.11 ± 1.91 *^†^	6.90 ± 0.47 ^†‡^
HOMA-IR score	0.26 ± 0.20	11.30 ± 1.50 *	3.86 ± 0.66 *^†^	7.68 ± 0.68 *^†^	2.60 ± 0.24 *^†‡§^
HR (beats/min)	349.7 ± 24.7	344.8 ± 4.8	342.0 ± 20.1	330.3 ± 11.9	328.7 ± 49.1
SBP (mmHg)	101.0 ± 14.2	108.2 ± 11.3	99.1 ± 4.6	105.3 ± 8.5	110.0 ± 10.0
DBP (mmHg)	73.67 ± 12.74	79.40 ± 6.47	77.14 ± 5.37	82.67 ± 14.19	78.30 ± 5.77
cfPWV (m/s)	0.43 ± 0.02	0.56 ± 0.05 *	0.38 ± 0.02 ^†^	0.40 ± 0.01 ^†^	0.26 ± 0.03 *^†‡§^
p-Akt/t-Akt (A.U.)	1.59 ± 0.64	0.40 ± 0.10 *	0.67 ± 0.12	1.02 ± 0.08	1.88 ± 0.41 ^†§^
p-eNOS/t-eNOS (A.U.)	1.38 ± 0.21	0.76 ± 0.09 *	1.39 ± 0.16 ^†‡^	0.93 ± 0.05 *	1.55 ± 0.18 ^†‡^
Arterial NOx levels (μmol/μg protein)	14.8 ± 2.8	7.4 ± 1.8 *	24.8 ± 4.0 *^†^	19.2 ± 2.6 ^†^	30.2 ± 2.1 *^†‡§^
Average food intake (g/d)	20.30 ± 0.19	26.97 ± 0.56 *^§^	20.70 ± 0.40	25.80 ± 0.29 *^§^	26.42 ± 0.37 *^§^

CS, citrate synthase; HOMA-IR, homeostatic model assessment for insulin resistance; HR, heart rate; SBP, systolic blood pressure; DBP, diastolic blood pressure; cfPWV, carotid-femoral pulse wave velocity; p-Akt/t-Akt, phosphorylated-Akt/total-Akt; p-eNOS/t-eNOS, phosphorylated-eNOS/total-eNOS; NOx, nitrate/nitrite; Healthy, healthy-sedentary control group; OBESE-SED, OBESE-sedentary control group; OBESE-ET, OBESE-aerobic exercise training group; OBESE-CH, OBESE-Chlorella intake group; OBESE-ET+CH, OBESE-aerobic exercise training and Chlorella intake group. * *p* < 0.05 vs. Healthy. † *p* < 0.05 vs. OBESE-SED. ‡ *p* < 0.05 vs. OBESE-CH. § *p* < 0.05 vs. OBESE-ET. Values are mean ± standard error.

**Table 2 nutrients-16-03080-t002:** Comparison of animal characteristics compared with OBESE-SED group.

	Healthy (*n* = 6)	OBESE		
		ET (*n* = 6)	CH (*n* = 6)	ET+CH (*n* = 6)
	*p*-Value	*p*-Value	*p*-Value	*p*-Value
Body weight (g)	0.0003	0.0015	0.4577	0.5962
Epididymal fat mass (g)	0.0003	0.0014	0.1449	0.0001
Soleus muscle mass (g)	0.1749	0.0917	0.8448	0.6941
Soleus CS enzyme activity (µmol/g/min)	0.7576	0.0395	0.3280	0.0001
Fasting blood glucose levels (mmol/L)	0.0001	0.0001	0.0221	0.0001
HOMA-IR score	0.0001	0.0001	0.0003	0.0001
HR (beats/min)	0.8203	0.1492	0.4134	0.3242
SBP (mmHg)	0.3203	0.1864	0.3847	0.7949
DBP (mmHg)	0.2223	0.6877	0.6458	0.7602
cfPWV (m/s)	0.3370	0.8475	0.0463	0.0119
p-Akt/t-Akt (A.U.)	0.0954	0.1204	0.0006	0.0057
p-eNOS/t-eNOS (A.U.)	0.0206	0.0059	0.1105	0.0027
Arterial NOx levels (μmol/μg protein)	0.0481	0.0026	0.0039	0.0001
Average food intake (g/d)	0.0001	0.0001	0.0936	0.4317

CS, citrate synthase; HOMA-IR, homeostatic model assessment for insulin resistance; HR, heart rate; SBP, systolic blood pressure; DBP, diastolic blood pressure; cfPWV, carotid-femoral pulse wave velocity; p-Akt/t-Akt, phosphorylated-Akt/total-Akt; p-eNOS/t-eNOS, phosphorylated-eNOS/total-eNOS; NOx, nitrate/nitrite; Healthy, healthy-sedentary control group; OBESE-SED, OBESE-sedentary control group; OBESE-ET, OBESE-aerobic exercise training group; OBESE-CH, OBESE-Chlorella intake group; OBESE-ET+CH, OBESE-aerobic exercise training and Chlorella intake group.

## Data Availability

The data presented in this study are available upon request (ethical reason) from the corresponding author.

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
