# Peer review of "Combined Aerobic Exercise Training and Chlorella Intake Reduces Arterial Stiffness through Enhanced Arterial Nitric Oxide Production in Obese Rats"

_nutrients, 2024, doi:10.3390/nu16183080_

Round 1
Reviewer 1 Report
Comments and Suggestions for Authors
The article by Yamazaki et al aims to examine the interaction between exercise training (treadmill 1 hr/day, 5 days/week, 8 weeks) and chlorella (0.5%, supplemented in food) on arterial stiffness through nitric oxide production. The authors use a LETO control group and OLETF rats (n=6/group) in 4 of the following treatment groups: sedentary, treadmill, chlorella sedentary, and chlorella exercise This is an interesting study looking at this nutrient supplementation on health in obesity. This group has previously published on the OLETF model with chlorella supplementation describing the impacts on glycemic control. and this reviewer has the following comments and concerns:
Given that this study is remarkably similar to the previous study, using the same methods, rat model, outcomes, yet slightly different n size (6 in this study, 7 in the previous), it is not clear if this is a different follow-up study or an analysis of tissue collected from the previous study. This must be clarified. If this is an additional analysis from tissue from the previous study the authors must be completely transparent regarding why the n size is different and why they are republishing some information that was published previously.
Further, study that examines food intake related to obesity, do the authors have data regarding food intake for each animal? Did the chlorella-treated animals eat less than the non-treated counterparts? Eating fewer calories can elicit critical changes in arterial stiffness. Did the authors track animal weights pre-, during, and post-study to determine the effects of diet on weight? Terminal body weight does not give this information.
It would be helpful to have individual data points plotted on the graph to evaluate the spread of the results.
The methods are incomplete. How were samples processed for western blotting? What method was performed for protein correction? Total protein loading in each well, which is the most appropriate method.
Chlorella is not a singular nutrient, yet contains lots of vitamins, minerals, amino acids, etc. Do the authors know the components of their chlorella? This is not addressed. Further, how many more calories does the chlorella add to the diet. This makes food intake even more important.
The authors conclude that chlorella further reduces arterial stiffness via activated arterial Akt/eNOS signaling, yet the effects of p-enos (Fig 2B) do not show this. The data suggests p-enos is driven by exercise alone.
The data does, however, show an increase in NOx levels with chlorella. How can this disconnect be explained?
The western blot images are incomplete. The authors only include 1 blot for each in supplemental material, which do not seem to match up with the representative images in the figures. Where are the remaining images? What about loading control images?
Author Response
RESPONSE TO REVIEWER COMMENTS
We thank all reviewers for their time and effort spent carefully reviewing our manuscript and providing constructive comments. We hope that the reviewers find the revised version of the manuscript satisfactory. Our point-by-point responses to all comments and suggested modifications are listed below.
Reviewer #1:
Major comments:
Comments 1: Given that this study is remarkably similar to the previous study, using the same methods, rat model, outcomes, yet slightly different n size (6 in this study, 7 in the previous), it is not clear if this is a different follow-up study or an analysis of tissue collected from the previous study. This must be clarified. If this is an additional analysis from tissue from the previous study the authors must be completely transparent regarding why the n size is different and why they are republishing some information that was published previously.
Response 1: Thank you for your comment. The aim of this study was different from that of our previous study (Horii N et al. Nutrition. 2019). In our previous study, we examined the effects of aerobic exercise and Chlorella intake on skeletal muscle energy metabolism. In this study, using the same study design and rat model, we examined the effects on arterial stiffness. However, we examined some of the same parameters, such as body weight, epididymal fat, and soleus muscle citrate synthase (CS) activity, as they are indicators for adaptation to aerobic exercise training.
Comments 2: Further, study that examines food intake related to obesity, do the authors have data regarding food intake for each animal? Did the chlorella-treated animals eat less than the non-treated counterparts? Eating fewer calories can elicit critical changes in arterial stiffness. Did the authors track animal weights pre-, during, and post-study to determine the effects of diet on weight? Terminal body weight does not give this information.
Response 2: Thank you very much for your comment. We have added data for the average food intake in each group to revised Table 1 and the Results section of the revised manuscript (page 4, lines 156-160). There were significant differences in average food intake between the OBESE-SED, OBESE-CH, and OBESE-ET+CH groups and the Healthy and OBESE-ET groups. However, there were no significant differences in average food intake among the OBESE-SED, OBESE-CH, and OBESE-ET+CH groups, suggesting that this parameter was not related to arterial stiffness. We did not track animal weights before and during the study.
Comments 3: It would be helpful to have individual data points plotted on the graph to evaluate the spread of the results.
Response 3: Thank you for your suggestion. We have replaced the graph with a box plot that includes individual data points to better visualize the spread of the results.
Comments 4: The methods are incomplete. How were samples processed for western blotting? What method was performed for protein correction? Total protein loading in each well, which is the most appropriate method.
Response 4: Thank you for your question. Following your suggestion, we have included more detailed descriptions of the methods for western blotting in the Methods section of revised manuscript (page 3, lines 101-106).
Comments 5: Chlorella is not a singular nutrient, yet contains lots of vitamins, minerals, amino acids, etc. Do the authors know the components of their chlorella? This is not addressed. Further, how many more calories does the chlorella add to the diet. This makes food intake even more important.
Response 5: Thank you for your comment. We did not include data for the major components of Chlorella because this information was reported in our previous study (Horii N. et al. Nutrition. 2019). As described in the Methods section, the OBESE-CH and OBESE-ET+CH groups were fed the same food supplemented with only 0.5% Chlorella powder. Thus, the calories of diet in the OBESE-CH and OBESE-ET+CH groups were similar to those in the normal diet administered to the other groups. Additionally, no significant difference in average food intake was observed among the OBESE-SED, OBESE-CH, and OBESE-ET+CH groups
Comments 6: The authors conclude that chlorella further reduces arterial stiffness via activated arterial Akt/eNOS signaling, yet the effects of p-enos (Fig 2B) do not show this. The data suggests p-enos is driven by exercise alone. The data does, however, show an increase in NOx levels with chlorella. How can this disconnect be explained?
Response 6: Thank you for your comment. There are several potential explanations for these findings. First, it has been reported that arterial NO production is related only to eNOS but also to neuronal NOS (nNOS) (Roy R et al. Int J Mol Sci. 2023). Therefore, the Chlorella-induced increase in arterial NO production might be regulated by nNOS in arterial vessels. Second, eNOS and nNOS are expressed not only in blood vessels but also in other tissues (Roy R et al. Int J Mol Sci. 2023). Thus, it is possible that Chlorella-induced increases in eNOS and nNOS in tissues other than the artery contributed to the increase in arterial NO production through blood vessels. However, we did not measure arterial nNOS or eNOS and nNOS in tissues other than the artery. We have identification this as a limitation in the Discussion section of the revised manuscript (page 9, lines 269-273).
Comments 7: The western blot images are incomplete. The authors only include 1 blot for each in supplemental material, which do not seem to match up with the representative images in the figures. Where are the remaining images? What about loading control images?
Response 7: Thank you for your attention to detail. We have added all western blot images to the supplemental material. In accordance with the guidelines of Nutrients, we have marked the samples not used in the analysis with a red 'X' (indicating that the excluded samples were from tissues other than the aorta and were used in a preliminary experiment).
Reviewer 2 Report
Comments and Suggestions for Authors
Firstly, and in my opinion, the manuscript addresses an intriguing topic concerning the influence of physical activity, specifically aerobic exercise.
I believe that the presentation of the objective needs improvement (lines 55-59). Scientific writing should be economical in word usage; thus, I suggest that the authors revise the objective to unify the two aims. Specifically, they should clarify whether the habitual intake of Chlorella combined with aerobic training influences... (enumerate the two aspects that the research intends to investigate).
The statistical method used (ANOVA) is appropriate; however, the limited sample size should be considered as it may introduce significant bias in the interpretation of the results. I recommend that the authors address the following:
Standard Deviation Reporting: The standard deviation should be reported, as this measure provides a more accurate representation of dispersion in this context. I have some concerns and would like to see the estimates of the standard deviation.
Median Data: I would also like to know the median values, as any extreme values could significantly affect the interpretation of the results.
Additionally, although the variance analysis is correctly performed, I recommend that the authors include a table comparing means or medians, as appropriate.
Subsequently, they should conduct pairwise comparisons of each group’s results with the SED group, which serves as the reference group. If means are used, a t-test should be performed; if medians are used, a non-parametric test should be applied.
Author Response
RESPONSE TO REVIEWER COMMENTS
We thank all reviewers for their time and effort spent carefully reviewing our manuscript and providing constructive comments. We hope that the reviewers find the revised version of the manuscript satisfactory. Our point-by-point responses to all comments and suggested modifications are listed below.
Reviewer #2:
Major comments:
Comments 1: I believe that the presentation of the objective needs improvement (lines 55-59). Scientific writing should be economical in word usage; thus, I suggest that the authors revise the objective to unify the two aims. Specifically, they should clarify whether the habitual intake of Chlorella combined with aerobic training influences... (enumerate the two aspects that the research intends to investigate).
Response 1: Thank you for your suggestion. We have unified the two the aims in the Introduction section of the revised manuscript (page 2, lines 57-58).
Comments 2: The statistical method used (ANOVA) is appropriate; however, the limited sample size should be considered as it may introduce significant bias in the interpretation of the results. I recommend that the authors address the following:
Response 2: Thank you for your comment. Following your suggestion, we have revised the Figures and Tables to address all points regarding statistical analyses.
Comments 3: Standard Deviation Reporting: The standard deviation should be reported, as this measure provides a more accurate representation of dispersion in this context. I have some concerns and would like to see the estimates of the standard deviation.
Response 3: Thank you for your suggestion. We have replaced the graph with a box plot that includes the standard deviation to better understand the dispersion.
Comments 4: Median Data: I would also like to know the median values, as any extreme values could significantly affect the interpretation of the results.
Response 4: Thank you for your suggestion. We have replaced all figures with box plots to display the results, including median values (represented by a line that splits the box into two equal sections).
Comments 5: Additionally, although the variance analysis is correctly performed, I recommend that the authors include a table comparing means or medians, as appropriate.
Response 5: Thank you for your suggestion. To address this point, in Table 1, we have added comparisons of mean values for all parameters included in figures. We have also replaced the figures with box plots, marking the mean values with an “X.”
Comments 6: Subsequently, they should conduct pairwise comparisons of each group’s results with the SED group, which serves as the reference group. If means are used, a t-test should be performed; if medians are used, a non-parametric test should be applied.
Response 6: Thank you for your suggestion. We have shown the mean values for all parameters in Table 1 and performed unpaired t-tests for comparisons between each group (Healthy, OBESE-ET, OBESE-CH, and OBESE-ET+CH groups) and the OBESE-SED group; we have added the results of these statistical analyses to Table 2 of the revised manuscript.
Round 2
Reviewer 2 Report
Comments and Suggestions for Authors
am reporting favorably for publication. The authors have made the modifications that I suggested in my previous report.